# Extra-Pseudocapsular Transsphenoidal Surgery for Microprolactinoma in Women

**DOI:** 10.3390/jcm11133920

**Published:** 2022-07-05

**Authors:** Juan Chen, Xiang Guo, Zhuangzhuang Miao, Zhuo Zhang, Shengwen Liu, Xueyan Wan, Kai Shu, Yan Yang, Ting Lei

**Affiliations:** 1Department of Neurosurgery, Tongji Hospital, Tongji Medical College, Huazhong University of Science and Technology, Wuhan 430030, China; jchen@tjh.tjmu.edu.cn (J.C.); d202082008@hust.edu.cn (X.G.); zzmiao@tjh.tjmu.edu.cn (Z.M.); zhangzhuo@tjh.tjmu.edu.cn (Z.Z.); 2011tj0534@hust.edu.cn (S.L.); xywan@tjh.tjmu.edu.cn (X.W.); kshu@tjh.tjmu.edu.cn (K.S.); 2Sino-German Neuro-Oncology Molecular Laboratory, Tongji Hospital, Tongji Medical College, Huazhong University of Science and Technology, Wuhan 430030, China; 3Department of Endocrinology, Tongji Hospital, Tongji Medical College, Huazhong University of Science and Technology, Wuhan 430030, China; yangyan@tjh.tjmu.edu.cn

**Keywords:** extra-pseudocapsule, transsphenoidal surgery, microprolactinoma, women

## Abstract

A recall for histological pseudocapsule (PS) and reappraisal of transsphenoidal surgery (TSS) as a viable alternative to dopamine agonists in the treatment algorithm of prolactinomas are getting vibrant. We hope to investigate the effectiveness and risks of extra-pseudocapsular transsphenoidal surgery (EPTSS) for young women with microprolactinoma, and to look into the factors that influenced remission and recurrence, and thus to figure out the possible indication shift for primary TSS. We proposed a new classification method of microprolactinoma based on the relationship between tumor and pituitary position, which can be divided into hypo-pituitary, para-pituitary and supra-pituitary groups. We retrospectively analyzed 133 patients of women (<50 yr) with microprolactinoma (≤10 mm) who underwent EPTSS in a tertiary center. PS were identified in 113 (84.96%) microadenomas intraoperatively. The long-term surgical cure rate was 88.2%, and the comprehensive remission rate was 95.8% in total. There was no severe or permanent complication, and the surgical morbidity rate was 4.5%. The recurrence rate with over 5 years of follow-up was 9.2%, and a lot lower for the tumors in the complete PS group (0) and hypo-pituitary group (2.1%). Use of the extra-pseudocapsule dissection in microprolactinoma resulted in a good chance of increasing the surgical remission without increasing the risk of CSF leakage or endocrine deficits. First-line EPTSS may offer a greater opportunity of long-term cure for young female patients with microprolactinoma of hypo-pituitary located and Knosp grade 0-II.

## 1. Introduction

Prolactin (PRL)-secreting adenomas are the most common type of pituitary adenomas. About 90% of prolactinomas are ≤1 cm and they occur predominantly in young women (aged 20 to 50 years old) [1,2]. Currently, dopamine agonists (DAs) are highly effective in achieving the goals to normalize prolactin and ameliorate the symptoms of hyperprolactinemia [3,4]. While most cases respond well to DAs, the drugs are often needed for many years. Moreover, the recurrence rates of hyperprolactinemia after withdrawal of medical therapy achieved 26–64% [5,6,7,8]. In contrast, transsphenoidal surgery (TSS) for prolactinomas is only accepted as a second-line treatment, recommended to symptomatic patients who cannot tolerate high dose of DA or are resistant to it, although the published surgical rates of PRL level normalization in some centers, especially for microprolactinoma, are excellent, or at least comparable with medical therapy [9,10,11,12]. Thus, long-term medication for young female patients, as the “gold-standard” treatment, is arguable nowadays [13,14].

The histological pseudocapsule (PS) associated with pituitary tumors was noted several decades ago, but few authors have examined its use in prolactinomas [12]. Therefore, our purpose in the current study was to analyze our surgical series in order to confirm the safety and efficacy of extra-pseudocapsular transsphenoidal surgery (EPTSS) for young female patients with microprolactinoma. We also focused on the factors influencing the surgical remission, as well as recurrence, and further elucidated the possible indication shift for primary transsphenoidal adenomectomy of microprolactinoma.

## 2. Materials and Methods

### 2.1. Study Population and Design

The authors prospectively enrolled consecutive young female patients (<50 years old) with microprolactinoma (≤10 mm) who underwent extra-pseudocapsular transsphenoidal surgery (EPTSS) from January 2012 to December 2015 (Figure 1A). Routine follow-ups were carried out until the end of July 2021 via phone or out-patient clinic. Ethics Committee approval was obtained from the Research Ethics Committee of Huazhong University of Science and Technology, People’s Republic of China. Informed consent was obtained from all patients after full explanation of the purpose and the nature of all the procedures.

The diagnosis of microprolactinoma was established via the patient’s symptoms, serum PRL level and preoperative magnetic resonance imaging (MRI) scanning. Most of the young females complained of menstrual problems, accompanied/not by lactation, infertility, hypolibido and headache. Normal PRL levels were considered to be 5–25 ng/mL for women at the Department of Laboratory Medicine, Tongji Hospital. The serum PRL level of the patient was significantly higher than the normal upper limit and defined as hyperprolactinemia. MRI examination can further identify the pituitary adenoma and distinguish it from other pituitary space-occupying lesions. When serum prolactin level was lower than 100 ng/mL, differential diagnoses include the intake of various drugs, compression of the pituitary stalk by other pathology (such as Rathke’s cyst, craniopharyngioma), hypophysitis, hypothyroidism, renal failure, cirrhosis or idiopathic hyperprolactinemia. All surgical intervention needs to be seriously discussed with endocrinologists and/or gynecologists, and alternative medical treatment with DAs should be mentioned to the patients. Specimen pathology testing confirmed PRL immunohistochemical staining was positive. Notably, cases of multiple hormone-secreting adenomas (such as PRL and GH, or PRL and TSH) were excluded.

### 2.2. Classification of Microprolactinomas

An original classification was used on the basis of the location relationship between the pituitary and tumor, which was first proposed by the senior surgeon (T.L.). The authors found out the PS is more observable and evident at the interface between the tumor and pituitary. In order to implement extra-pseudocapsular adenomectomy, the strategy was to aim at the interface according to the location relationship and to identify the PS first after opening the dura. Pituitary adenomas arise from a single cell within the anterior pituitary. With the tumor growing slowly, the adjacent normal gland is squeezed, resulting in a surgical pseudocapsule that encases the entire adenoma. The tumor–pituitary locational relationship is highly related to which sides of hypophysis the initial tumor cells originated from (Figure 2A). Accordingly, small intrasellar tumors can be subdivided into three groups. (1) In the hypo-pituitary group, the neoplasm originates from the anterior wall or the floor of the sella turcica and pushes the hypophysis tissue posteriorly and superiorly. Subsequently, most of the tumor is beneath the pituitary, sitting at the bottom of sella. (2) In the para-pituitary group, the tumors initiate from one lateral side of pituitary and extend to the medial wall of the cavernous sinus (CS). In the meantime, they push the pituitary to the other side. (3) In the supra-pituitary group, the prior tumor cells are mainly located close to the dorsum sella or suprasellar side, along with the pituitary stalk. The tumor cells grow superiorly and posteriorly to the suprasellar region, giving rise to the normal pituitary localizing at the bottom of the sella. The coronal view of MRI with contrast images demonstrated this classification clearly (Figure 2B). For larger tumors extending out of sellar region, this locational relationship was obscured, since the pituitary tissue was elongated and became too thin to be classified. In addition to the novel classification, Knosp grading was also employed to qualify the growth pattern and the CS invasiveness of pituitary adenomas [15].

### 2.3. Extra-Pseudocapsular Transsphenoidal Surgery (EPTSS) for Pituitary Adenomas

Surgical indications for patients in this study included five aspects (Table 1). Assisted by neuro-navigation and neuro-endoscopy, all patients underwent a trans-septal sphenoidal approach to the sellar, and in the extra-pseudocapsular tumor resection manner, mainly performed by the senior author (L.T.), who is an experienced pituitary surgeon with over 350 TSSs per year. Firstly, the double-layer dura of the sellar floor was incised broadly enough to expose the pituitary–tumor interface. The compressed anterior pituitary gland was characterized by its orange color, firmer consistency and vascular surface structure, and was carefully preserved. The microdissector (2.8 mm Hardy, Codman) was employed to separate the border at the interface between the pituitary gland and the tumor. In order to remove the tumor en bloc, the pseudocapsule plane was gently detached and preserved from the pituitary side, then to other surrounding parts.

During the growth of microprolactinoma, normal pituitary tissue was squeezed and displaced by the tumor, resulting in the formation of an interface between the tumor and normal pituitary composed of compressed normal glands. Because normal glands contain collagen and mesh proteins, a pseudocapsule formed around the tumor.

All adenomas were qualitatively classified in accordance with the existence and integrity of pseudocapsule as follows: (1) “complete” group, those with a distinct pseudocapsule that are easily removed along the whole tumor mass, and (2) “incomplete” group, which cannot be dissected en bloc with part of indistinct PS, (3) cases in which those without any residual of PS tissue were discovered under microscopy or endoscopy were ascribed to “no-capsule” group.

### 2.4. Remission and Recurrence Criteria

Anterior pituitary function was closely assessed and monitored by means of symptoms, signs and endocrine tests, including morning fasting basal hormone levels Outpatient follow-up was carried out after one month, three months, then yearly ever after. MRI studies were performed before the surgery and three months after surgery. MRI follow-up underwent annually. Gross total resection (GTR) was defined as no significant residual tumor on MRI 3 months postoperatively.

Initial remission was considered to have occurred if prolactin levels were within the normal range on the first day postoperatively. In contrast, a significant decrease in prolactin levels on the first day after surgery without a return to normal was defined as partial remission. Twelve weeks after surgery is the earliest time point for the evaluation of surgical remission, which was achieved when the patient had resolution of symptoms, disappearance of tumor mass (certified by gross total resection on MRI), plus normal PRL levels.

Recurrence was considered to have occurred for surgical remission patients if the symptoms came back during follow-up and increased prolactin levels without the MRI finding tumor reoccurrence. Long-term remission was defined by the latest follow-up before August 2021 with the absence of hyperprolactinemia or DA medication.

### 2.5. Statistical Analysis

We used descriptive statistics as well as contingency coefficients (CCs) and standardized mean differences (SMDs) to indicate the sample size–independent magnitude of group differences. Pearson’s chi-square test, Fisher’s exact test, two-sample *t*-test or a Mann–Whitney U-test were used depending on the scaling and distribution of the variables. Time for detection of tumor enlargement and median recurrence-free survival times were estimated using the Kaplan–Meier method. Odds ratios (ORs) and 95% confidence intervals (CIs) of independent factors for surgical remission positive outcome were analyzed using univariable and multivariable logistic regression. Results with *p* < 0.05 were considered statistically significant. All statistical analyses were performed using SPSS Statistics, version 21 (IBM Corp., Armonk, NY, USA).

## 3. Results

### 3.1. Clinical Characteristics

One hundred and thirty-three young women were eligible for analysis. The mean preoperative PRL level was 146.19 ± 103.84 ng/mL (extreme values: 37.64–518.83 ng/mL), 25 patients (18.8%) greater than 200 ng/mL. DAs (bromocriptine (the only CFDA permitted prescribed DA in China)) were primarily used on 44 patients (33.1%). Postoperatively, 10 patients (7.5%) still had symptoms related to hyperprolactinemia; 4 patients’ menstrual disorder did not relieve although the nomarprolactinemia was reached. In addition, postoperative tumor pathology showed Ki-67 ≥ 3% in 23 cases (17.3%) (Table 2). 

### 3.2. Pseudocapsule Status in the Microprolactinoma

Intraoperatively, PS tissue was found in 113 tumors (85.0%) (Table 3). In 53 (46.9%) tumors, the PS was visualized as a well-developed capsule entirely covering the tumor mass; for 60 (53.1%) tumors, the membrane was discontinuously developed (Table 4). The initial remission rate and surgical remission rate are both higher in PS complete group (96.2% vs. 90.0%, 94.2% vs. 86.3%), but are not statistically significant (*p* > 0.05).

In 62.7% hypo-pituitary prolactinomas, the microsurgical pseudocapsule tended to exist more prominently. Conversely, only one from the supra-pituitary group developed a complete capsule and the majority (90%) were more difficult to manipulate surgically (*p* < 0.05). To predict the completeness of the PS in microprolactinoma, ROC curve analysis showed the novel classification of area under the curve was slightly higher than the Knosp grade (Figure 2D). Additionally, 60% of supra-pituitary microprolactinomas and 50% (one in two cases) of Grade IV of Knosp grading found no PS in EPTSSs (Table 4). Furthermore, the results showed this group of patients had a smaller tumor diameter (7.4 ± 2.0 vs. 8.6 ± 1.6 mm), higher postoperative PRL level (*p* < 0.05), and lower postoperative remission rate.

### 3.3. Early Surgical Remission and Associated Factors

The median postoperative PRL concentration on the first day after surgery was 1.98 ng/mL (0.25–55.81 ng/mL), the average postoperative PRL was 6.16 ± 10.10 ng/mL and 7.55 ± 13.68 ng/mL, respectively on the 1st day and 12th week. A total of 127 patients achieved total tumor resection (95.5%). As shown in Figure 1, on the 12th week multidimension evaluation, 119 patients (89.5%) achieved surgical remission. Four patients had hyperprolactinemia relapses, but no residue was found on MRI images.

For the tumors with a well-developed capsule, 50/53 (94.3%) patients reached the surgical remission, compared with only 86.3% for incomplete and no PS group, but the difference has no statistical significance. The surgical remission rate was 97.2% (105/108) for patients with preoperative PRL level lower than 200 ng/mL, while women with PRL values of ≥200 ng/mL experienced poorer outcomes (56.0% remission 14/25). Univariable analysis revealed that lower preoperative PRL level, lower Knosp grade, hypo-pituitary group, pseudocapsule complete and GTR were related to positive surgical remission results. Furthermore, the preoperative PRL level was noted in the multivariable analysis (Table 4 and Table 5).

### 3.4. Postoperative Complications

No perioperative mortality or major morbidity occurred in our cohort. A total of six patients developed postoperative complications of different types and degrees, and the overall incidence rate of postoperative complications was 4.5% (Table 6). There were two patients who experienced severe epistaxis and required emergent tamponade. One patient (0.8%) was observed with CSF rhinorrhea in hospital, and relieved after lumbar cistern drainage. Within the first week postoperatively, five patients (3.8%) had transient diabetic insipidus, three required medication management, and none required additional intervention after discharge. No SIADH case was recorded.

There were 10 patients who developed anterior pituitary hypofunction before surgery, including 4 patients with complete PS and another 6 patients in the incomplete group. Overall, two patients who did not improve after surgery had incomplete/no PS. Two additional patients had newly onset of declined hormone level without presenting severe symptoms and signs of hypopituitarism. Comparably, in the PS complete group, all the patients with injured pituitary endocrine function returned to normal range after surgery; additionally, no new episode was found.

### 3.5. Long-Term Follow-Up and Recurrence

After a three-month outpatient follow-up, 14 patients were lost to contact in the most recent evaluations. Thus, data from 119 patients with long-term follow-up ranging from 3 to 101 months were available (68.9 months on average). Overall, 11 out of 119 remission patients were confirmed to have a relapse of their hyperprolactinemia or tumor recurrence after a median period of 24 months (extreme values: 11–42 months). Of these 11 patients, only 1 patient resulted in relapse of hyperprolactinemia within the first year after surgery. Exceeding 5 years of follow-up, 102 (85.7%) patients achieved long-term remission, defined as surgical cure. When we included another 3 patients who received DA postoperatively and reached complete withdrawal, a total of 105 patients (88.2%) achieved long-term DA free cure. The 5-year disease free survival rate was 88.2% for the whole group of patients.

In the PS complete group, 45 patients (93.8%) reached long-term remission, except for a single patient who had recurrent (2.1%) with hyperprolactinemia 13 months after surgery. The Kaplan–Meier survival curve was used to compare the recurrence time and recurrence rate of patients in PS complete and incomplete/no groups. The results showed that the long-term recurrence rate of the PS complete group was significantly lower (*p* < 0.05). The 5-year disease-free survival rate was 97.9% in the PS complete group and 85.9% in the PS incomplete/no group (Figure 2G).

Of hypo-pituitary group (Table 7), 42 patients (95.5%) reached long-term remission and none experienced recurrence (*p* < 0.05). The Kaplan–Meier survival curve also revealed the five-year recurrence rate in hypo-pituitary group was zero. The 5-year disease-free survival rate was 84.8% in the para-pituitary group and 88.9% in the supra-pituitary group (Figure 2H). 

Univariable analysis revealed that preoperative medication of DA, higher preoperative PRL level, higher Knosp grades (Grade 3 and 4), non-hypo-pituitary type, no/incomplete pseudocapsule and sub-GTR were related to postoperative recurrence outcomes. Preoperative DA treatment, higher preoperative PRL level and sub-GTR for postoperative recurrence were also noted in the multivariable analysis (Table 4 and Table 5). Figure 2F demonstrate that for the long-term recurrence rate in this series, the diagnostic significance of the novel classification was slightly higher than the Knosp grading. The diagnostic value of two combined for evaluation was more significant.

## 4. Discussion

The present study is the first formal analysis of pseudocapsule status in microprolactinoma. The results suggest that TSS intervention in an extra-pseudocapsular mode for microprolactinoma is associated with an excellent surgical cure rate of 88.2%, and a long-term remission rate of 95.8% in total. There were no severe or permanent complications, and the total surgical morbidity rate was 4.5%. These data were comparable to most other published series. Furthermore, the recurrence rate in our cohort with more than 5 years of follow-up is 9.2%, which is setting in the lower range compared with the previous published series for surgical intervention of microprolactinomas [13,16,17,18]. We also present that the surgical outcome was much better particularly in the complete PS, hypo-pituitary and Knosp grade 0-II groups.

The term “pseudocapsule” was initially introduced by Costello in 1936, identified at autopsy and described as “the compressed normal gland at the edge of an adenoma” [12]. Until 2006, Oldfield and Vortmeyer described the histology of pseudocapsule in Cushing’s disease, and the landmark technical operation of en bloc extracapsular dissection was well illustrated [19]. Use of the histological pseudocapsule as a surgical capsule has been previously demonstrated to improve gross total resection, remission rates and rate of tumor recurrence, as well as to improve the resolution of preoperative endocrinopathies among microadenomas [20,21,22]. Utilization of the pseudocapsule among prolactinomas, however, has been discussed only in limited series and with little to no emphasis on surgical technique and efficacy. This may ascribe to the limited reported surgical cases of prolactinomas. Our surgical team first recognized the existence and importance of pseudocapsule in pituitary surgeries in 2011 and introduced EPTSS for all categories of adenomas. In this series of microprolactinomas, the PS detection rate was achieved at 84.96%, which is much higher than the published data [20]. This may ascribe to the large surgery volume and the growing experience. Neuroendoscopy has also played an important role for surgeons to observe closer and clearly. Additionally, this might be associated with previous medical treatment with dopamine agonists for PRL-secreting adenomas, which facilitate fibrous tissue formation. However, the results from this study shows preoperative medication is not a predictor of PS status.

The operative strategy of EPTSS is to actively discover the pseudocapsule plane forwards to remove the tumor. Our novel classification helps surgeons to identify the interface and manipulate it easily intraoperatively. The surgical technique of keeping this fragile membrane intact is also crucial; a major part of the PS is tender enough and can be held by the microtumor forceps (2 mm). In addition, preoperative knowledge of PS consistency affords the neurosurgeon substantial benefit. For most primary surgeries, the PSs press loosely to the surrounding and can be easily dissected if the right plane is found. On the contrary, for the tumors after long-term bromocriptine treatment with obvious shrinkage, the texture can become too firm to detach gently, resulting in sharp dissection, especially for the tumors breaking through the medial wall of cavernous sinus. These conditions may lead to incomplete or even no removal of the PS. In cases of larger or more invasive tumors, the consistency could be too thin or soft to grasp and remove.

On the other hand, concerns were raised from extending resection associated with the risk of hypopituitarism and CSF leak. In our surgical series, it was low at 4.5% and may not be related to the extracapsular mode. Disturbance of posterior pituitary is slightly high in 3.8% of patients; transient symptoms were all relieved before discharge. In contrast, excluding hyperprolactinemia, decompression of the pituitary gland by selective transsphenoidal adenomectomy restored pituitary functions in 6% of patients. With the employment of the surgical plane, normal pituitary tissue may be under better protection, using cotton-plat during dissection. Complication rates are low in centers with dedicated pituitary surgeons, as outcomes depend on surgical expertise, tumor size, and invasiveness [14,23,24]. Extra-pseudocapsule dissection also has obvious disadvantages. The operation is expected to take an average of 15 min longer than traditional procedures due to more elaborate and meticulous manipulation. The preoperative DA administration varied diversely in our cohort; 66.9% patients (due to personal will or cystic prolactinoma) received primary operations. We found that preoperative DA treatment is not a predictor of PS status or the surgical remission, but a negative factor for recurrence. Landolt and Osterwalder reported that patients treated with bromocriptine before surgery were significantly less likely to normalize prolactin due to perivascular and tumor fibrosis [24,25]. After reviewing surgical notes, we found the PS texture of DA-treated tumor was mostly firm and closely adhered to the surrounding tissue, especially to the medial wall of the cavernous sinus. In contrast, other analyses have noted improved outcomes when patients were treated with bromocriptine before surgery [10,26,27,28,29]. Whether therapy with a dopamine agonist exerts a negative effect on surgical outcome remains controversial. In addition to the classical indications for acute complications (tumor stroke or CSF rhinorrhea), drug intolerance or poor drug effects, tumor enlargement, and symptom aggravation during pregnancy, surgery is also a very important treatment option for young women who have not given birth, patients who are unwilling to accept long-term drug therapy and patients with excessive drug therapy [11].

The high recurrence rate for prolactinomas after TSS is the main arguable point in favor of primary surgical intervention [30,31]. Previous studies reported that the long-term recurrence rate of TSS varies from 0–58%. A recent large retrospective review found a long-term remission rate of 84.5% in microprolactinomas, and a recurrence rate of 18.7% overall using the microscope [24]. A high recurrence rate of hyperprolactinemia (34%) was also observed, occurring throughout a prolonged mean postoperative follow-up period of 94 months [32]. Our experience shows that the recurrence rate can be lowered by the pseudocapsule excision. Several studies had proved the tumor cells usually invade into the pseudocapsule with a detection rate of 30–51%, depending on the characteristics of the pituitary adenoma, including the pathological type and tumor size [20,33]. Notably, even in a tumor with a well-developed PS, tumor cells might invade the nearby normal pituitary gland, the dural layer beyond the PS and the medial wall of the cavernous sinus [34,35]. For this reason, Laws and Kuwayama et al. emphasized that it is necessary to remove a rim of normal pituitary gland surrounding the tumor to improve surgical outcome in patients with endocrine-active pituitary adenomas. However, this fact does not diminish the usefulness of pseudocapsule-based extracapsular resection.

With regard to another main intention of our study, namely a potential refinement of the current interdisciplinary guidelines from a neurosurgical standpoint, we designed our study to determine the potentially useful criteria for identifying patients with the highest likelihood of long-term remission after surgical therapy. The long-term remission rate is 95.5% for the group of hypo-hypophysis located adenomas, in which the most suitable cases are those surrounding the normal hypophysis tissue without lateral or superior invasion. Additionally, a lower preoperative PRL level was also a strong predictor of a surgically achieved long-term remission of hyperprolactinemia in our study, with a lower recurrence rate. Previously, a correlation of higher surgical remission rate with preoperative prolactin levels was demonstrated in several studies [36,37,38]. Para-hypophysis-located cases are closely related to the relationship of the medial wall of CS. In Knosp type II and type III, some tumors push the medial border of the CS aside so that the membrane is intact, while the others breakthrough inside the CS, giving rise to residual and easily relapse. Supra-pituitary prolactinomas are the most challenging cohort to find that surgical plane. We do not think the PS does not exist for these tumors, but it is quite difficult to identify and to keep whole during surgery using microscopy or endoscopy. After all, tumors with cavernous sinus, dural, or bone invasion are difficult to cure surgically [11,32].

Both successful antitumor and antisecretory efficacies of DA made them the first-line treatment of prolactinomas. Two recent meta-analysis confirmed surgery reaches better long-term remission compared to the DA withdrawal groups [13,24]. According to studies by Jethwa et al. and Zygourakis et al., the economic costs of surgical resection are comparable to those of pharmacological management over a 10-year period [39,40]. This suggests that beyond 10 years, surgery (performed by experienced surgeons with excellent cure rates and minimal rates of complications) may be a more cost-effective option than life-long medical therapy, especially in young patients. This explains the rising number of people who are reluctant to maintain long-term medical therapy and prefer a surgical alternative [11,16,23,26,40,41,42,43]. Another reason that we favor young female patients was that even transient remission could take advantage of the remission phase to become pregnant without having to resort to DA treatment. The relatively high number of patients resistant or intolerant to dopamine agonists is likely to reflect a referral bias to a center specialized in pituitary surgery.

There were several limitations to this study. First, a relatively small number of participants were included because surgery is not routinely performed in prolactinoma patients. Second, this was an observational study, and we only focused on the surgical outcomes in one center. Therefore, we must emphasize that these findings cannot be applied universally to all neurosurgical institutes. Furthermore, the specific indications for surgery may have introduced a bias in the selection of patients studied; the impact of this (positive or negative) in the reported outcomes is not clear. Clinically, a bias for microprolactinomas without engulfing of internal carotid artery and for those aged under 50 was already set up. Conversely, it is possible that the population of patients most amenable to surgical success (microadenomas) may also be the population most likely to achieve long-term remission after discontinuation of dopamine agonist treatment. Additional well-designed case–control studies should be performed to confirm the efficacy of EPTSS in conjunction with medical therapy for microprolactinoma.

## 5. Conclusions

Extra-pseudocapsule dissection is a viable and effective treatment option for microprolactinoma. This method increased the chances of surgical remission without increasing the risk of postoperative CSF leakage or endocrine impairments. Adenomas with complete PS resection may have a significant influence on postoperative care, resulting in lower recurrence rates. For young female patients with microprolactinoma of hypo-pituitary location and Knosp grade 0-II, first-line TSS in an extra-pseudocapsular modality may give a better chance of long-term cure. Surgery, on the other hand, should always be reserved as a last resort following dopamine agonist routine therapy for supra-pituitary or highly cavernous sinus invasive microprolactinoma.

## Figures and Tables

**Figure 1 jcm-11-03920-f001:**
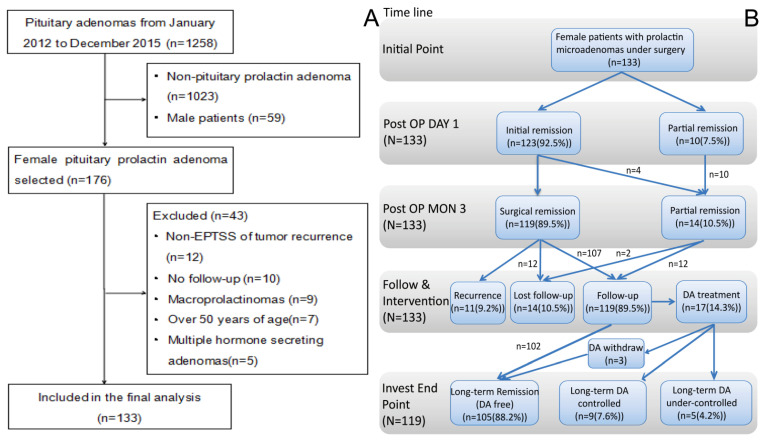
(**A**) The patient selection process is depicted in a flow chart. Surgery was conducted on 176 female pituitary prolactin adenoma patients out of 1258 patients with pituitary adenoma, with 133 patients included in the final study due to the presence of complete data. (**B**) Flowchart of treatment and prognosis of 133 female patients with microprolactinoma. All 133 female patients with microprolactinoma underwent transsphenoidal microsurgery to resect the tumor. Out of 123 (92.5%) patients who had initial remission, 10 (7.5%) patients had partial remission on the first day postoperatively. Three months after surgery, 119 (89.5%) patients achieved surgical remission, whereas 14 (10.5%) patients were in partial remission. A total of 14 (10.5%) patients were lost to follow-up, while 11 (9.2%) developed tumor recurrence during subsequent follow-up. Additionally, 102 (85.7%) of the 119 (89.5%) patients who were followed up on achieved long-term remission (DA free), 17 (14.3%) were treated with dopamine agonists (DA), 3 (17.6%) achieved remission after discontinuing DA therapy, 9 (53.0%) required long-term DA control and 5 (29.4%) were under-controlled with long-term DA therapy. EPTSS—extra-pseudocapsular transsphenoidal surgery; DA—dopamine agonists.

**Figure 2 jcm-11-03920-f002:**
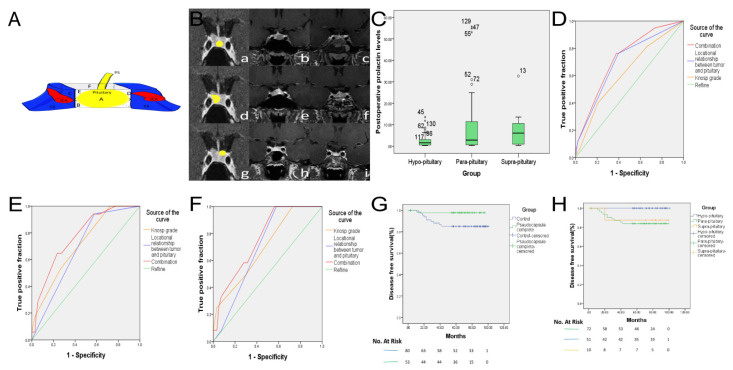
(**A**) PS—pituitary stalk; ICA—internal carotid artery; CS—cavernous sinus; side A—anterial wall of sella; side B—sellar floor; side C—lateral left side; side D—lateral right side; side E—sellar dorsum; side F—supra sellar. This is an anatomic pattern of the sellar region showing the pituitary gland, the pituitary stalk, the bilateral cavernous sinus and the internal carotid artery. We compare the pituitary gland to a hexahedron and try to describe and explain that the pituitary adenoma extension and development are highly related to which side of the pituitary the tumor cells initially originated from. Microprolactinoma are classified into hypo-pituitary type, para-pituitary type and supra-pituitary type. Microprolactinoma originating from both sides of A and B are mostly hypo-pituitary type, those originating from both sides of C and D are para-pituitary type, and those originating from both sides of E and F are mostly supra-pituitary type. (**B**) The model of the locational relationship between tumor and pituitary, and preoperative and postoperative MRI. b, c, e, f, h and i, MRI DFOV: 8.45 × 6.74 cm. a—hypo-pituitary, the tumor was located in the lower part of the pituitary and there was pituitary in the lateral wall of the cavernous sinus on both sides; b—preoperative MRI enhancement of the pituitary demonstrates that the tumor is located below the pituitary; d—para-pituitary, the tumor was located on one side of the pituitary, connected to the ipsilateral cavernous sinus, without pituitary; e—preoperative MRI enhancement of the pituitary showed that the tumor was located on the right side of the pituitary; g—supra-pituitary, the tumor is located above the pituitary; h—preoperative MRI enhancement of the pituitary showed that the tumor was located in the upper left of the pituitary; c, f and i, postoperative MRI enhancement of the pituitary showed that the tumor was completely removed and the pituitary structure remained intact. (**C**) Postoperative PRL levels in 133 female patients with different types of microprolactinoma. The prolactin levels of the hypo-pituitary type after operation were significantly lower than other two types. ° represent extreme values, which are values greater than three times the interquartile spacing. * represent outliers, which are values greater than 1.5 quartile spacing. (**D**) In this cohort, ROC analysis revealed that the locational relationship between tumor and pituitary had a greater predictive effect than the Knosp grade on whether the pseudocapsule was complete. Its area under the ROC curve was 0.698 (95% CI: 0.607~0.790). Additionally, combining the above two evaluation methods will have higher predictive value, its area under the ROC curve was 0.725 (95% CI: 0.636~0.814). (**E**) The Knosp grade for female microprolactinoma had a greater predictive effect on long-term remission rate than the locational relationship between the tumor and pituitary, according to ROC analysis. Its area under the ROC curve was 0.713 (95% CI: 0.593~0.834). Additionally, the above two evaluation methods combined have higher predictive value; the area under the ROC curve was 0.773 (95% CI: 0.664~0.882). (**F**) ROC analysis revealed that the locational relationship between tumor and pituitary had a more significant predictive effect on recurrence rate than the Knosp grade for female pituitary prolactin microadenomas. Its area under the ROC curve was 0.698 (95% CI: 0.587~0.808). Additionally, combining the above two evaluation methods’ higher predictive value, its area under the ROC curve was 0.761 (95% CI: 0.644~0.878). (**G**) Kaplan–Meier analysis of postoperative tumor recurrence time in 133 female patients with microprolactinoma, according to whether the pseudocapsule was complete. Control group represents pseudocapsule incomplete and no pseudocapsule group. (**H**) Kaplan–Meier analysis of postoperative tumor recurrence time in 133 female patients with microprolactinoma, according to the locational relationship between tumor and pituitary.

**Table 1 jcm-11-03920-t001:** Surgical indications for 133 female patients with microprolactinoma.

Surgical Indication	Patients (*n* = 133)
Drug intolerant (%)	8 (6.0)
Drug resistance (%)	36 (27.1)
Personal willing (%)	67 (50.4)
Cystic prolactinoma (%)	14 (10.5)
Tumor stroke (%)	8 (6.0)

**Table 2 jcm-11-03920-t002:** Baseline characteristics of 133 female patients with microprolactinoma.

Variables		Patients (*n* = 133)	
Age, years (mean ± SD)		27.3 ± 6.5	
Tumor diameter, mm (mean ± SD)		8.4 ± 1.7	
Ki-67 (%)			
<3		110 (82.7)	
≥3		23 (17.3)	
Follow-up time, months (median)		72	
PRL level, ng/mL (mean ± SD)		146.19 ± 103.84	
Clinical symptoms (%)	Preoperation	Postoperation	*p* value
Menstrual disorder	105 (78.9)	7 (5.3)	0.000 ^a^
Hyposexuality	7 (5.3)	1 (0.8)	0.066 ^b^
Galactosis	46 (34.6)	4 (3.0)	0.000 ^a^
Infertility	19 (14.3)	3 (2.3)	0.001 ^a^

^a^—χ^2^ test; ^b^—Fisher’s exact test.

**Table 3 jcm-11-03920-t003:** Baseline characteristics of pseudocapsule classification in 133 female patients with pituitary prolactin microadenoma.

	Total	Pseudocapsule Complete	Pseudocapsule Incomplete/No Pseudocapsule	*p* Value
Total (*n*)	133	53	80	
Age, years (mean ± SD)	27.3 ± 6.5	28.1 ± 6.8	26.8 ± 6.3	0.252 ^a^
Tumor diameter, mm (mean ± SD)	8.4 ± 1.7	8.7 ± 1.5	8.3 ± 1.8	0.202 ^a^
Ki-67 (%)				0.138 ^b^
<3	110 (82.7)	47 (88.7)	63 (78.8)	
≥3	23 (17.3)	6 (11.3)	17 (21.2)	
Preoperation PRL (ng/mL)				0.986 ^b^
<200	108 (81.2)	43 (81.1)	65 (81.3)	
≥200	25 (18.8)	10 (18.9)	15 (18.7)	
Preoperative medication of DA				0.564 ^b^
With	44 (33.1)	16 (30.2)	28 (35.0)	
Without	89 (66.9)	37 (69.8)	52 (65.0)	
Knosp grade (%)				0.000 ^c^
Grade 0	33 (24.8)	18 (34.0)	15 (18.8)	
Grade I	57 (42.9)	24 (45.3)	33 (41.3)	
Grade II	36 (27.1)	9 (17.0)	27 (33.8)	
Grade III	6 (4.5)	2 (3.8)	4 (5.0)	
Grade IV	1 (0.7)	0 (0)	1 (1.3)	
Locational relationship between tumor and pituitary (%)				0.001 ^c^
Hypo-pituitary	51 (38.4)	32 (60.4)	19 (23.7)	
Para-pituitary	72 (54.1)	20 (37.7)	52 (65.0)	
Supra-pituitary	10 (7.5)	1 (1.9)	9 (11.3)	
Follow-up time, months (mean ± SD)	61.9 ± 32.0	61.7 ± 28.7	62.1 ± 34.2	0.946 ^a^
Follow-up loss rate (%)	14 (10.5)	5 (9.4)	9 (11.3)	0.738 ^b^
Initial remission rate (%)	123 (92.5)	51 (96.2)	72 (90.0)	0.314 ^d^
Surgical remission rate (%)	119 (89.5)	50 (94.3)	69 (86.3)	0.137 ^b^
Long-term remission rate (%)	102 (85.7)	45 (93.8)	57 (80.3)	0.039 ^b^
Recurrence rate (%)	11 (9.2)	1 (2.1)	10 (14.1)	0.048 ^d^

^a^—t test; ^b^—χ^2^ test; ^c^—nonparametric Wilcoxon rank sum test; ^d^—Fisher’s exact test.

**Table 4 jcm-11-03920-t004:** Baseline characteristics of 20 female patients with lack of PS.

	Pseudocapsule Complete/Incomplete	No Pseudocapsule	*p* Value
Total (*n*)	113	20	
Age, years (mean ± SD)	27.3 ± 6.5	27.4 ± 6.6	0.928 ^a^
Tumor diameter, mm (mean ± SD)	8.6 ± 1.6	7.4 ± 2.0	0.018 ^a^
Ki-67 (%)			0.750 ^b^
<3	94 (83.2)	16 (80.0)	
≥3	19 (16.8)	4 (20.0)	
Knosp grade (%)			0.023 ^c^
Grade 0	27 (23.9)	3 (15.0)	
Grade I	53 (46.9)	4 (20.0)	
Grade II	27 (23.9)	9 (45.0)	
Grade III	5 (4.4)	3 (15.0)	
Grade IV	1 (0.9)	1 (5.0)	
Preoperative medication of DA			0.476 ^d^
With	36 (31.9)	8 (40.0)	
Without	77 (68.1)	12 (60.0)	
Locational relationship between tumor and pituitary (%)			0.000 ^c^
Hypo-pituitary	47 (41.6)	4 (20.0)	
Para-pituitary	62 (54.9)	10 (50.0)	
Supra-pituitary	4 (3.5)	6 (30.0)	
Preoperation PRL, ng/mL (mean ± SD)	150.14 ± 106.11	123.90 ± 88.98	0.299 ^a^
Postoperation PRL, ng/mL (mean ± SD)	4.81 ± 8.74	13.81 ± 13.64	0.009 ^a^

^a^—t test; ^b^—Fisher’s exact test; ^c^—nonparametric Wilcoxon rank sum test; ^d^—χ^2^ test.

**Table 5 jcm-11-03920-t005:** Predictors of surgical remission positive outcome and postoperative recurrence.

Predictive	Uni Variable Analyses OR (95% CI)	*p* Value	Multivariable Analyses OR (95% CI)	*p* Value
Age	1.060 (0.980–1.147)	0.148		
	1.078 (0.963–1.207)	0.191		
Medication (preoperative)	1.357 (0.539–3.416)	0.517		
	4.020 (1.142–14.151)	**0.030**	8.075 (1.309–49.813)	**0.024**
Tumor diameter	1.200 (0.946–1.521)	0.133		
	0.915 (0.618–1.356)	0.659		
Ki-67	0.923 (0.709–1.202)	0.553		
	1.001 (0.659–1.520)	0.997		
Preoperative PRL levels	0.993 (0.989–0.996)	**0.000**	0.992 (0.986–0.998)	**0.015**
	0.986 (0.981–0.992)	**0.000**	0.982 (0.970–0.93)	**0.002**
Postoperative PRL levels	0.807 (0.751–0.867)	**0.000**	0.830 (0.758–0.908)	**0.000**
	0.827 (0.762–0.897)	**0.000**	0.831 (0.752–0.918)	**0.000**
Knosp grading	0.466 (0.274–0.790)	**0.005**	0.981 (0.404–2.378)	0.966
	0.322 (0.155–0.669)	**0.002**	0.444 (0.087–2.265)	0.329
Locational relationship between tumor and pituitary	0.449 (0.217–0.932)	**0.032**	1.546 (0.420–5.684)	0.512
	0.319 (0.118–0.861)	**0.024**	0.492 (0.045–5.425)	0.563
Classification of pseudocapsule	2.759 (1.436–5.302)	**0.002**	1.819 (0.595–5.556)	0.294
	3.134 (1.286–7.638)	**0.012**	0.419 (0.025–6.956)	0.544
GTR	25.682 (8.470–77.873)	**0.000**	1.901 (0.297–12.170)	0.498
	62.400 (14.346–271.423)	**0.000**	0.018 (0.003–0.110)	**0.000**

CI—confidence intervals; OR—odds ratio; PRL—prolactin; GTR—gross total resection.

**Table 6 jcm-11-03920-t006:** Postoperative complications and evaluation of hormonal involvement preoperation and post operation.

Variables		Patients (*n* = 133)	
Postoperative complications (%)			
Epistaxis		2 (1.5)	
CSF rhinorrhea		1 (0.8)	
Temporary diabetes insipidus		5 (3.8)	
Hypophysis hypofunction		4 (3.0)	
Hormonal axis	Preoperation	Postoperation	*p* value
Adrenal axis (%)	5 (3.8)	2 (1.5)	0.447 ^a^
Gonadal axis (%)	2 (1.5)	1 (0.8)	1.000 ^a^
Thyroidal axis (%)	3 (2.3)	1 (0.8)	0.622 ^a^

^a^—Fisher’s exact test.

**Table 7 jcm-11-03920-t007:** Effect of treatment and prognosis for 133 female patients with microprolactinoma, according to the locational relationship between tumor and pituitary.

	Hypo-Pituitary	Para-Pituitary	Supra-Pituitary	*p* Value
Total (*n*)	51	72	10	
Age, years (mean ± SD)	27.8 ± 7.2	27.5 ± 6.2	23.4 ± 3.6	0.143 ^a^
Tumor diameter, mm (mean ± SD)	8.6 ± 1.6	8.5 ± 1.6	7.3 ± 2.3	0.087 ^a^
Ki-67(%)				0.589 ^b^
<3	44 (86.3)	57 (79.2)	9 (90.0)	
≥3	7 (13.7)	15 (20.8)	1 (10.0)	
Preoperation PRL (ng/mL)				0.083 ^b^
<200	46 (90.2)	54 (75.0)	8 (80.0)	
≥200	5 (9.8)	18 (25.0)	2 (20.0)	
Knosp grade (%)				0.000 ^c^
Grade 0	27 (52.9)	4 (5.6)	2 (20.0)	
Grade I	18 (35.3)	35 (48.6)	4 (40.0)	
Grade II	4 (7.8)	28 (38.9)	4 (40.0)	
Grade III	2 (3.9)	4 (5.6)	0 (0)	
Grade IV	0 (0)	1 (1.4)	0 (0)	
Classification of pseudocapsule (%)				0.000 ^c^
Complete	32 (62.7)	20 (27.8)	1 (10.0)	
Incomplete	15 (29.5)	42 (58.3)	3 (30.0)	
No	4 (7.8)	10 (13.9)	6 (60.0)	
Preoperation PRL, ng/mL (mean ± SD)	125.06 ± 68.52	162.25 ± 123.87	138.40 ± 79.72	0.143 ^a^
Postoperation PRL, ng/mL (mean ± SD)	2.51 ± 2.99	8.45 ± 12.48	8.33 ± 9.76	0.004 ^a^
Initial remission rate (%)	51 (100)	63 (87.5)	9 (90.0)	0.015 ^b^
Surgical remission rate (%)	50 (98.0)	61 (84.7)	8 (80.0)	0.018 ^b^
Long-term remission rate (%)	42 (95.5)	53 (80.3)	7 (77.8)	0.045 ^b^
Recurrence rate (%)	0 (0)	10 (15.2)	1 (11.1)	0.012 ^b^

PRL—prolactin; ^a^—one-way ANOVA; ^b^—Fisher’s exact test; ^c^—nonparametric Kruskal–Wallis H(K) test.

## Data Availability

The data that support the findings of this study are included in the article. Further inquiries are available from the corresponding author upon reasonable request.

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
