# Peer review of "Extra-Pseudocapsular Transsphenoidal Surgery for Microprolactinoma in Women"

_jcm, 2022, doi:10.3390/jcm11133920_

Round 1
Reviewer 1 Report
We carefully reviewed this manuscript.
The contribution is very interesting and the authors reported a quite large series.
The topic is very interesting and the analysis is well conducted.
We would like the authors to take in consideration several criticism in the attempt of improving the readability of the paper.
- The concept of pseudo-capsule should be detailed as seem little distant form the Oldfield description for ACTH secreting tumors.
- Manuscript is really too long and should be shortened.
- Series should not include tumors extending into cavernous sinus or supradiaphragmatic space.
- The classification is not clear and it seems to have poor clinical and surgical implications; please clarify this topic and eventually discuss the outcomes in regards to tumor location.
- Indications for surgery in prolactinoma is not discussed and need to be addressed.
- References needs to be updated and upgraded
- Remission and recurrence predicting factors are not clear and should better depicted.
Reviewer 2 Report
The best therapy for prolactinomas – dopamine-agonists (DA) vs. surgery – is still strongly debated. In this context, the authors discuss the presence of a pseudocapsule, and the effect of extracapsular transsphenoidal surgery (TSS) on surgical outcome in a cohort of 133 microprolactinomas in women. Extracapsular macroprolactinoma removal was association with favorable outcome. It is the first publication with detailed analysis of pseudocapsules in prolactinomas.
In addition, a new classification depending on tumor location and its significance for outcome is presented. The categories of location are “hypo-pituitary”, “para-pituitary” and “supra-pituitary”. “Para-pituitary” macroprolactinoma site was associated with very low recurrence rate. The new classification is interesting and clinically useful.
The study design is clear and straightforward. The article is clearly structured, fulfills formal requirements, and is logically organized. The initial question is clearly elaborated and adequately answered in the paper after detailed analysis. The discussion is well written and includes the relevant literature in depth.
The study is clearly of interest for the readers.
The specific points of criticism and recommendations are addressed in the following specific comments:
· Abstract: “hypo-pituitary” is not understandable when reading the abstract alone. It might be confused with hypo-pituitarism. I suggest that the new classification (which is a main issue of the manuscript) with the categories “hypo-pituitary”, “para-pituitary” and “supra-pituitary” is introduced to the readers in the abstract.
· I recommend to use “microprolactinoma” throughout the manuscript. Different uncommon terms are used (f.e. “pituitary prolactin adenoma”).
· Material and Methods: The term “partial remission” needs to be defined and specified.
· Lines 276-277: The use of primary DA has to be specified. It could be described under Material and Methods. When were DA discontinued prior to surgery? If DA were taken until the date of surgery, they would influence the initial remission rate. Were DA discontinued immediately after surgery in those patients with remission after 3 months?
· Line 39: Some publications mention higher recurrence rates after withdrawal of DA.
· Lines 69-70: The sentence “Further MRI …” should be re-phrased. The wording is cumbersome and not correct.
· Figure 1 B: It is difficult to understand the line “Follow&Intervention”. The numbers also do not add up to N=133. The Figure could be improved.
· Line 321: A “histological PS” is mentioned. If the PS is not always proved by histo-pathology, “histological” should be deleted. If the PS was always confirmed by histo-pathology, it should be described under Methods.
· Table 4 is unclear for me. Behind single predictive variables, several analysed variables appear. How is it meant? Some horizonal lines could help. The Table should be re-done.
· Lines 453-454: Please check the numbers. 93.8% and 2.1% do not add up to 100%.
· Table 6 has the wrong heading.
· Line 540: Regarding the increase in duration of operations, a more precise temporal specification would be interesting (see line 540).
· The proposed new classification should be emphasized in the “discussion”.
· Since this is a study on microprolactinomas, the unit in line 323 should be mm and not cm, if applicable.
· In line 444 a range of 65 - 101 months is described with an average of 61.9 months. There might be a number error here.
· Lines 584-585: It should not be expressed categorically “… cannot be cured surgically.”
The English language is easily understandable. However, the English wording and grammar is not always correct and there are some typos. Thorough proofreading by a native speaker would improve the English language.
